# Identification, Classification and Characterization of LBD Transcription Factor Family Genes in *Pinus massoniana*

**DOI:** 10.3390/ijms232113215

**Published:** 2022-10-30

**Authors:** Chi Zhang, Peihuang Zhu, Mengyang Zhang, Zichen Huang, Agassin Romaric Hippolyte, Yangqing Hou, Xuan Lou, Kongshu Ji

**Affiliations:** Key Laboratory of Forestry Genetics & Biotechnology of Ministry of Education, Co-Innovation Center for Sustainable Forestry in Southern China, Nanjing Forestry University, Nanjing 210037, China

**Keywords:** lateral organ boundaries domain, *Pinus massoniana*, gene expression, pine wood nematode

## Abstract

Transcription factors (TFs) are a class of proteins that play an important regulatory role in controlling the expression of plant target genes by interacting with downstream regulatory genes. The lateral organ boundary (LOB) structural domain (*LBD*) genes are a family of genes encoding plant-specific transcription factors that play important roles in regulating plant growth and development, nutrient metabolism, and environmental stresses. However, the *LBD* gene family has not been systematically identified in *Pinus massoniana*, one of the most important conifers in southern China. Therefore, in this study, we combined cell biology and bioinformatics approaches to identify the *LBD* gene family of *P. massoniana* by systematic gene structure and functional evolutionary analysis. We obtained 47 *LBD* gene family members, and all *PmLBD* members can be divided into two subfamilies, (Class I and Class II). By treating the plants with abiotic stress and growth hormone, etc., under qPCR-based analysis, we found that the expression of *PmLBD* genes was regulated by growth hormone and abiotic stress treatments, and thus this gene family in growth and development may be actively involved in plant growth and development and responses to adversity stress, etc. By subcellular localization analysis, PmLBD is a nuclear protein, and two of the genes, *PmLBD44* and *PmLBD45*, were selected for functional characterization; secondly, yeast self-activation analysis showed that *PmLBD44*, *PmLBD45*, *PmLBD46* and *PmLBD47* had no self-activating activity. This study lays the foundation for an in-depth study of the role of the *LBD* gene family in other physiological activities of *P. massoniana*.

## 1. Introduction

Transcription factors (TFs) regulate the expression of target genes in plants by selectively binding to the promoters of eukaryotic genes. LOB structural domain (LBD) proteins are characterized as proteins with a conserved lateral organ boundary (LOB) domain [1]. LBD proteins are crucial TFs that control the development of plant organs, but they also have a range of other roles during the growth and development of plants.

An N-terminal domain that is largely conserved and a variable C-terminal section make up the LBD protein. The LOB structural domain is part of the N-terminal region and contains the zinc finger-like motif (CX2CX6CX3C) for DNA binding activity, the gas blocking motif (Gly-Ala-Ser), and the leucine zipper-like coiled-coil motif (LX6LX3LX6L) for protein dimerization [2]. Conserved proline residues in the GAS region were discovered to be crucial to the biological activities of *Arabidopsis* LBD proteins, and the C-terminal region confers transcriptional activation/repression of target gene expression. The *LBD* gene family members were split into two subfamilies based on sequence similarity and phylogenetic study (Class I and Class II). Class I LBD proteins can be divided into five branches (IA, IB, IC, ID, and IE) if they contain zinc finger-like motifs, gas clusters, and leucine zipper-like helical motifs, whereas class II LBD proteins without complete leucine zipper-like structural domains can be divided into two branches (IIA and IIB) [3].

Class I *LBD* gene use is primarily engaged in plant development, such as lateral root, leaf, and flower development, per the research thus far, which has demonstrated that the *LBD* gene family plays a vital role in diverse regions of plant growth, development, and metabolism [4]. Class II *LBD* genes, on the other hand, might be engaged in metabolic procedures like the formation of anthocyanins and extra nitrogen reactions. *AtLBD4* and the *AtLBD41*, a gene discovered by Chalfun-Junior et al. [5] are important in controlling plant leaf development in *Arabidopsis*. The *AtAS2* gene, another member of the *LBD*, is selectively expressed on the proximal surface of developing floral organs, controlling the development of these organ in plants [6,7]. Scheible et al. [8] and Rubin et al. [9] found that class Ⅱ *LBD* members *AtLBD37*, *AtLBD38*, *AtLBD39* in *Arabidopsis* and the *OsLBD37* gene found in rice by Albinsky et al. [10] are involved in the regulation of nitrogen metabolism. Zentella et al. [11] found that gibberellin inhibits the expression of *AtASL37*, a member of the *LBD*; while Ikezaki et al. [12] found that another member of the *LBD*, the *AtAS2* gene, was able to promote gibberellin synthesis. Berckmans et al. [13] found that *Arabidopsis AtLBD33* and *AtLBD18* genes were able to promote lateral root development by activating the expression of *E2Fa*. The *LBD1* gene, which is homologous to *Arabidopsis AtLBD1* in poplar, is able to fuse with the repressor SRDX structural domain translation, reduce diameter growth, inhibit silique development, and regulate secondary growth [14].The *AtLBD20* gene, a root-specific *LBD* gene in *Arabidopsis*, is a negative regulator of susceptibility to root infection with the fungus pathogen *Fusarium*, in disease resistance [15,16]. The most crucial gene for citrus’s defense against *AtLBD1* and *AtLBD11* is *CsLOB1*, which is the closest citrus homolog to those two genes. Citrus suffers from bacterial ulcer disease due to a broad susceptibility gene caused by a range of *Xanthomonas* species [17,18]. Additionally, *AtLBD16* is a well-known regulator of Aux-triggered lateral root development that especially encourages the growth of the large cells and galls that make up the feeding structures for root-knot nematodes (RKNs) [19,20]. Multiple pathogens stimulate nearly all class II *LBD* genes, according to the expression profiles of members of the *LBD* gene family, demonstrating the role of *LBD* genes in plant defensive reactions [16]. In conclusion, because *LBD* genes transcriptionally control a variety of downstream genes, they are important molecular targets for plant pathogen invasion. The *LBD* gene family has been identified and researched in a number of species [3], including poplar [21], tomato [22], barley [23], potato [24], moso bamboo [25], and *Populus trichocarpa* [26], but *P. massoniana* has not yet been the subject of such research.

A member of the Pinaceae family and an important timber species in southern China, *P**. massoniana*, is also a pioneer species for the regeneration of arid mountains [27]. It is a significant timber for forestry and industrial production, used mostly for building, paneling, furniture, and wood fiber products. Pine resin, a raw ingredient for drugs and chemicals, can be extracted from the trunk by cutting it. Throughout its life cycle, *P. massoniana* is vulnerable to biotic and abiotic stressors. Therefore, it is necessary to study and transform abiotic stress genes to improve the growth quality of horsetail pine. The primary culprit of pine wood nematode(PWN) disease is the pine wood nematode (*Bursaphelenchus xylophilus*), which causes pine trees to wilt quickly and has a complex pathogenic process that is still poorly understood [28].

The *P. massoniana LBD* gene family have not been discovered because of its genome’s enormous size. Therefore, using its transcriptome datasets [29], we were able to identify the *LBD* family members. By phylogenetic grouping and protein motif structure analysis, the *PmLBD* family was categorized using the *Arabidopsis LBD* family classification as a guide [30]. The *PmLBD* family was classified by phylogenetic grouping and protein motif structure analysis. In addition, the basic information, physicochemical properties, genetic structure, evolutionary relationships, conserved structural domain features and expression patterns of *P. massoniana LBD* genes were comprehensively analyzed [29]. This study lays the foundation for future studies on *LBD* genes in *P. massoniana*, and also facilitates future studies on the response of *P. massoniana* to environmental stresses.

## 2. Results

### 2.1. Identification and Phylogenetic Grouping of LBD Family Proteins in P. massoniana

We identified a total of 47 putative full-length *LBD* family genes (*PmLBD1*-*PmLBD47*) from *P. massoniana* (Appendix A). The protein sequence lengths of the *PmLBD* genes ranged from 58 to 362 amino acids; the molecular masses ranged from 6.58 to 39.86 kDa; the isoelectric points ranged from 4.49 to 9.72; the instability coefficients ranged from 41.61 to 76.75, and all *LBD* gene family members had stability coefficients greater than 40, which were unstable proteins; aliphatic amino acid indexes were 56.09~100.41; hydrophilic indexes were −0.591~0.171 (Appendix A).

### 2.2. Phylogenetic Analysis and Classification of the PmLBD Gene Family 

#### 2.2.1. Construction of a Phylogenetic Tree of the *PmLBD* Gene Family

In order to deeply analyze the evolutionary relationship between *PmLBD* genes and *Arabidopsis LBD* genes, we selected several typical *Arabidopsis LBD* members of each subclass with *P. massoniana* to construct an *LBD* phylogenetic tree with reference to the grouping method of *Arabidopsis LBD* gene families [31] (Figure 1). The evolutionary tree clustering results showed that Class Ⅰ was subdivided into five subclasses: ⅠA, ⅠB, ⅠC, ⅠD, and ⅠE, and *P. massoniana* contained 20, 9, 1, 1, and 9 *LBD* family members, respectively. Class Ⅱ was subdivided into two subclasses ⅡA and ⅡB, and *P. massoniana* contained three and four *LBD* family members, respectively. The Class I subfamily contains 40 members containing the complete and highly conserved cysteine structural domain CX2CX6CX3C, the GA_S_ motif and the leucine-like zipper motif LX6LX3LX6L; the Class Ⅱ subfamily contains seven members containing only the complete and highly conserved cysteine motif CX2CX6CX3C.

#### 2.2.2. Structural Prediction and Subcellular Localization of *PmLBD* Gene Family Proteins

In the results of protein secondary structure prediction and analysis, it was found that the secondary structures of five genes, *PmLBD9*, *PmLBD19*, *PmLBD25*, *PmLBD41* and *PmLBD47*, contained only two conformations, α-helix and irregular curl, while the rest of the genes contained three conformations. The distribution of α-helices in *LBD* genes ranged from 22.87% to 66.27%, and the distribution of irregular curls ranged from 25.43% to 58.91%, both of which were distributed in the protein sequences and accounted for a relatively large proportion (Table 1). The subcellular localization results indicated that the *PmLBD* genes were all localized in the nucleus (Appendix A). In addition, two genes were selected for transient expression experiments to further explore the subcellular localization characteristics of the *PmLBD* family members. By observing the position of GFP fluorescence and chloroplast autofluorescence of PBI121-PmLBD shown in the legend and the hybrid map showing that the two are not at the same site, plus comparing with the fluorescence position map of the empty load, we can basically judge that the gene is localized at the nucleus. The fluorescent signals of *PmLBD44* and *PmLBD45* were observed in the nucleus (Figure 2).

#### 2.2.3. Tertiary Structure Prediction of *PmLBD* Gene Family Proteins 

Based on the SWISS-MODEL database, homology modeling of PmLBD proteins was performed, and the structures of the members of the two subfamilies were predicted (Figure 3). The structures with the highest GMQE and QMean were selected as the best structures of PmLBD proteins. Therefore, *PmLBD47* from class Ⅰ and *PmLBD44* from class Ⅱ were selected for study. Each subfamily has two chains, A and B, and exhibits a symmetric “Y” structure. A similar “pocket” region is formed in the combination of A and B chains, and the model contains multiple α-helices with few irregular curls and β-turns, and no β-folding is found, indicating that the region is highly protected. At the same time, there are some conserved structures at the amino acid terminus and carboxyl terminus of each subfamily, and it can be inferred that the genes and functions of the two subfamilies of class Ⅰ and class Ⅱ are similar at the protein structure level.

### 2.3. Analysis of the Conserved Structural Domain of the PmLBD Gene Protein

The analysis of the number of conserved structural domains and their constituent positions in the characteristic regions of the 47 member proteins of the *PmLBD* gene family predicted 10 conserved motifs, and the specific amino acid sequences of the characteristic motifs (motif1, motif2, motif3) were listed (Figure 4). All of them obtained motif1, which is a zinc finger-like structure CX2CX6CX3C, which is also a hallmark of the *LBD* gene family. The majority of the members have both motif2 and motif3, and by comparison motif2 is a GAS motif and motif3 is a leucine-like zipper structure LX6LX3LX6L. Among them, motif 1 was identified in all *PmLBD* genes, indicating that motif 1 was the most conserved. Motifs 1 and 2 were identified in all Class Ⅱ subfamilies, and motif 1 was only present in group ⅠD, while motifs 4, 5, and 9 were only present in ⅠA, ⅠE, and ⅠB, respectively, indicating that most members of the same subfamily have the same sequence and motifs in common position, and *PmLBD* members clustered in the same subgroup may have similar biological functions (Figure 5).

### 2.4. Gene Expression Analysis by Transcriptome Data of PmLBD during Inoculation with Pine Nematodes

To assess the potential role of *PmLBD* on inoculated pine nematodes, we investigated gene expression patterns based on transcriptomic data (SRA accession: PRJNA66087). The results showed that two *PmLBD*s belonging to CLASS Ⅰ and two *PmLBD*s belonging to CLASS Ⅱ family were transcribed during inoculation with pine nematodes (Figure 6). The expression of most *LBD* family members was reduced after inoculation with pine nematodes, with the exception of *PmLBD46*. The abundance of most *PmLBD*s was lower at 10 day, and the expression level of *PmLBD46* was higher at 35 day.

### 2.5. Analysis of PmLBD Genes Based on qPCR under Different Treatments

The results in Figure 7 show the responses of four *PmLBD* genes to different abiotic stresses. Under ABA treatment (Figure 7a), *PmLBD46* and *PmLBD47* showed a positive response, while the remaining genes were not sensitive. All genes were induced significantly by NaCl treatment, especially *PmLBD47* (Figure 7b). The expression of all four *PmLBD* genes (*PmLBD44*, *PmLBD45*, *PmLBD46*, and *PmLBD47*) were significantly upregulated at 24 h under MeJA treatment (Figure 7c). Under SA treatment (Figure 7d), *PmLBD45*, *PmLBD46* and *PmLBD47* decreased first and increased after, while *PmLBD44* showed the opposite. After IAA treatment, only *PmLBD44* showed an up-regulation trend at 12 h (Figure 7e). Under PEG treatment (Figure 7f), the expression of *PmLBD45 and PmLBD47* increased first, reached the peak, and then decreased, and the expression of *PmLBD44*, and *PmLBD46* were significantly down-regulated at 3 h, 6 h, 12 h, and 24 h. After the mechanical injury (Figure 7g), *PmLBD44*, *PmLBD45* and *PmLBD47* showed a positive response, while the remaining gene *PmLBD46* was not sensitive to mechanical injury. Under H_2_O_2_ treatment, the expression of *PmLBD44* was significantly up-regulated at all time points while *PmLBD45* and *PmLBD46* only was significant at 24 h, especially *PmLBD45*. In addition, *PmLBD44* was not sensitive to H2O2 treatment (Figure 7h). *PmLBD44* was significantly upregulated at 12 h, while *PmLBD45*, *PmLBD46*, and *PmLBD47* became significantly downregulated at 3 h, 6 h, 12 h, and 24 h under ETH treatment (Figure 7i).

### 2.6. qPCR-Based Gene Analysis of PmLBD under Different Tissues

The expression of *PmLBD44*, *PmLBD45*, *PmLBD46*, and *PmLBD47* in different tissues of 15-year-old *P. massoniana* was analyzed using real-time fluorescence quantitative PCR. The results (Figure 8) showed that *PmLBD44*, *PmLBD45*, *PmLBD46*, and *PmLBD47* were expressed in roots, apices, young leaves, old leaves, young stems, old stems, xylem, and bark, but at different levels of expression. *PmLBD44* was significantly higher in roots than in other tissues, 2.7-fold and 3.4-fold higher than in young leaves and apices. The expression level of *PmLBD46* was significantly higher in older leaves than in other tissues, 3.9-fold and 2.1-fold higher than in young leaves and apices, so *PmLBD46* was mainly expressed in older leaves. In contrast, the expression levels of *PmLBD45* and *PmLBD47* were not significantly higher under each tissue compared with the young leaves and all other parts.

### 2.7. Carrier Toxicity Assay and Validation of Self-Activating Activity

In theory the binding domain (BD) vector can bind to the upstream activating sequence (UAS) alone, but cannot cause transcription. However, if a segment of a transcription factor with self-activating activity is constructed onto the BD vector, and its expression produces a decoy protein (bait) that can induce transcription of downstream reporter genes after binding to the UAS alone, it indicates that the fusion decoy protein has a self-activating phenomenon. Alternatively, some endogenous proteins of yeast cells with transcriptional activation activity may also interact with the decoy protein. This makes it impossible to determine whether it is due to bait interactions with prey proteins (prey). Therefore, only on the basis of excluding self-activation can we determine whether bait interacts with prey. In addition, in some cases, strains poorly cultured in liquid medium can grow well on solid medium, indicating that bait proteins are toxic to yeast cells. In this experiment, by picking yeast colonies containing bait carriers on SD/-Trp plates and yeast containing empty carriers after inoculation in SD/-Trp liquid medium, the OD values were measured separately after overnight incubation, and it was found that the growth trends of both had exceeded the standard value of 0.8, and then the self-activation was continued on SD/-Trp/-His/-Ada/X-α-gal for verification. A transcriptional self-activation assay was performed by observing the growth and growth status of yeast single colonies on nutrient-deficient medium. Yeast grew normally on SD/-Trp solid medium, indicating that the recombinant decoy protein was not toxic to the growth of *Saccharomyces cerevisiae.* AH109 could be assayed for self-activation activity. In contrast, four recombinant decoys, *PmLBD44*, *PmLBD45*, *PmLBD46* and *PmLBD47*, did not grow normally and did not show blue color on SD/-Trp/-His/-Ade/ X-α-gal solid medium (Figure 9). The vector transcriptional self-activation assay showed that *PmLBD44*, *PmLBD45*, *PmLBD46* and *PmLBD47* did not have transcriptional self-activation activity and were ready for the next step of hybridization assay to screen the proteins.

## 3. Discussion

*LBD* genes have been identified in many plant species, and different gene families play different roles in plants. Therefore, a detailed classification of *LBD* genes is crucial, which is important for gaining insight into the functions of *LBD* genes. Since there are multiple transcripts in the *LBD* genes of *P. massoniana*, for the case of multiple transcripts of the same gene, the last one transcript was selected as the representative sequence, and after excluding the sequences with missing and incomplete LOB structural domains, a total of 47 protein sequences were obtained and used as members of the *PmLBD* gene family. The phylogenetic tree of the *LBD* gene family of *P. massoniana* and *Arabidopsis* showed that all *PmLBD* genes were divided into two subclades (ClassⅠ, ClassⅡ). ClassⅠ (40, 85.1%) and ClassⅡ (7, 14.9%), in terms of gene structure and protein conserved motifs, and the structural differences between the two subclades of *PmLBD* genes were small, but the protein conserved motifs differed significantly. Similar to *Arabidopsis* [6], the number of class Ⅰ members was higher than class Ⅱ during the evolutionary process, and their phylogenetic relationships were essentially the same as those of previous studies [25,30]. Homology modeling of the members of the two subfamilies of PmLBD proteins showed that each subfamily consists of two chains, A and B, in a symmetrical “Y” structure and forms a similar “pocket” region at the junction. The “pocket” region is similar at the junction, indicating a high degree of conservation in this region. Structural analysis can provide valuable information on the phylogenetic relationships and evolutionary duplication events of gene members in the same gene family. Gene sequence and gene structure analyses have shown that closely related gene members often exhibit similar motif composition, as in *Arabidopsis* [5] and other plants such as rice [7]. Both interspecific and intraspecific phylogenetic trees als34o share a high degree of similarity, indicating that the *LBD* gene family is evolutionarily conserved.

Members of the class Ⅰ family are associated with plant development [32,33], while class Ⅱ family genes are important candidates for abiotic stress tolerance and biotic stress resistance in plants [34]. Advances in sequencing technologies and the availability of large data sets have helped to identify *LBD* family genes and to classify several plant species using genomic [35], transcriptomic [36] or EST data [37]. Due to the large genome size of the Pinaceae species, which is close to 20 GB in size, there is a lack of data on the molecular and genetic characteristics of it and limited access to genomic information. In recent years, only a few genomic databases have been established for *P**inus taeda* [38,39], *Pcea abies* [39], *P. glauca* [40], and *P. tabuliformis* [41]. The lack of reference genomic data limits the study of *P. massoniana*. Transcriptome sequencing is a feasible and economical technique for generating relatively comprehensive sequencing data in a short period of time, and the technique has become popular in plant research [42]. 

LBD-TFs are thought to be involved in the response to pine nematode disease [43]. Gene expression patterns are thought to be directly related to gene function, and in this study, based on transcriptome sequencing data (SRA accession: PRJNA66087), transcript abundances of four *PmLBD* genes were obtained during pine nematode disease resistance. The expression of all four genes was upregulated to some extent. In addition, based on qPCR analysis, all four selected genes showed significant up-regulation under MeJA treatment. MeJA, as a plant hormone and signaling molecule related to damage, is widely present in plant questions, and exogenous application can stimulate the expression of defense plant genes, which has been verified in the existing studies [16]. Therefore it is likely to be associated with pine nematode resistance in *P. massoniana*.

Furthermore, the expression of four *LBD* family genes from groups I and II differed depending on different stresses and growth hormones of treatment. The different expression patterns of genes in the same family suggest that the functional diversity of *LBD* genes is associated with different treatments. In addition, these family members also regulate gene expression in response to a range of biological stimuli, including insects as well as downstream defense signaling hormones such as SA, MeJA, ETH, and ABA. Furthermore, these significantly differentially expressed genes may be closely associated with the signaling process in response to treatment conditions as the treatment duration increases. In summary, the identification, classification, and expression profile of *PmLBD* family genes provide insight into the biological functions of individual LBD-TFs in *P. massoniana*. Meanwhile, it was demonstrated that *PmLBD44*, *PmLBD45*, *PmLBD46* and *PmLBD47* did not possess transcriptional self-activating activity by observing the growth and growth status of yeast containing the pGBKT7-PmLBD vector on nutrient-deficient medium (Figure 9), which could be used for subsequent yeast two-hybrid assays. Interaction validation can only proceed if the decoy vector is free of both self-activating activity and toxicity. To verify the nuclear localization, two genes, *PmLBD44* and *PmLBD45*, were randomly selected for subcellular localization in this study, and the results showed localization at the nucleus (Figure 2).

## 4. Materials and Methods

### 4.1. Identification of the Members of the LBD Gene Family of P. massoniana

To identify the members of the *LBD* gene family contained in the genome of *P. massoniana*, a model was first built based on the characteristic sequence of the LOB conserved structural domain (PF03195) provided by the database Pfam website, and then with the help of the Hidden Markov Model (HMM) in the local CO_2_-stressed transcriptome [44], the young branch transcriptome (SRA accession: PRJNA655997), the drought-stressed transcriptome (SRA accession: PRJNA595650) and the transcriptome of inoculated pine nematodes (SRA accession: PRJNA66087) in the FASTA file, and feature finding was performed for this transcription factor to obtain relevant data for all *LBD* gene family members contained in the Marestail pine transcriptome database. Screening was performed using HMMER3 (v3.0) software to identify *LBD* families in *P. massoniana* (its threshold value for definition was set to be less than 10^−5^). Additionally, with the aid of online identification algorithms from the SMART and HMMER websites, the candidate genes screened in the previous stage were once more ranked and determined to include LOB structural domains. The ineligible findings that lacked the LOB conserved structural domains were eliminated before the final sequences could be chosen.

### 4.2. Sequence Analysis, Subcellular Localization and Phylogenetic Tree Construction

Using MEGA 7.0 [45], many amino acid sequence comparisons were carried out, and LOB structural domain areas were chosen. Based on the sequence of the LOB structural domain of PmLBD, a phylogenetic tree was created. To compare and analyze the evolutionary relationships as well as subgroup types of both, the evolutionary tree of the AtLBD protein and the PmLBD protein was built using the Neighbour-Joining method in MEGA 7.0 software (setting Bootstrap value to 1000, Partial deletion value to 50, and other parameters as default values). Utilizing the internet tool EvolView (http://evolgenius.info//evolview-v2/, accessed on 19 July 2022), the evolutionary trees were enhanced. Using the ProtParam tool (https://web.expasy.org/ProtParam/, accessed on 19 July 2022), the proteins’ theoretical molecular weight, isoelectric point (pI), instability coefficient, aliphatic amino acid index, and hydrophilicity index were examined. SOPMA (http://npsa-prabi.ibcp.fr/cgi-bin/nps, accessed on 3 July 2022) was used to gather pertinent data and secondary structures. Using SWISS-MODEL (https://swissmodel.expasy.org/, accessed on 3 July 2022), an online program based on the homology modeling method, the tertiary structure of LBD proteins was predicted.

The CELLO program (http://CELLO.life.nctu.edu.tw/, accessed on 22 September 2022) and Plant-mPLoc were used to predict the subcellular localization of PmLBD proteins. For temporary conversion, *PmLBD44*, *PmLBD45*, *PmLBD46*, and *PmLBD47* were chosen, and the primers used to create the vector are listed in Appendix A. The green fluorescent protein (GFP) expression vector PBI121-GFP had the open reading frame (ORF) region added into it (GFP). After creating transient expression vectors (35S::PmLBD44-GFP, 35S::PmLBD45-GFP), the *Agrobacterium rhizogenes* strain GV3101 was transformed with these vectors. These two strains were cultivated in LB medium for 36 h at 28 °C with an additional p19 (RNA silencing blocker) *Agrobacterium* strain before being suspended in injection solution (10 mM MgCl_2_, 10 mM 2-(N-morpholinyl)ethanesulfonic acid (MES), 150 M acetosyringone). *N. benthamiana* plants that were 30 to 40 days old had their leaves injected with a 1:1 mixture of suspension cells and p19 for temporary transformation. The injected *N. benthamiana* plants were kept in a dark culture for 48 h at 22 °C in a growth chamber. Using an LSM710 confocal laser scanning microscope, GFP signals from *N. benthamiana* leaves were recorded.

### 4.3. Identification of PmLBD Proteins’ Conserved Structural Domains

To acquire conserved structural domain areas, DNAMAN 7.0 software was used to perform multiple sequence alignment on the sequences of the PmLBD proteins that were found throughout the screening process. MEME 5.1.1 (http://MEME-suite.org/tools/MEME, accessed on 8 July 2022) [46] was used to evaluate the amino acid sequences of LBD proteins with the following settings: The maximum number of motifs was 10, and the ideal width of the motifs was 20–60 amino acids. The *LBD* sequences’ conserved structural domain tags were created utilizing the WebLogo online platform (*http://weblogo.threeplusone.com/*, accessed on 8 July 2022).

### 4.4. Transcriptional Profiling of the PmLBD Gene

RNA sequencing (RNA-seq) data (SRA accession: PRJNA66087) from the transcriptome of PWN inoculated were acquired for the investigation of *PmLBD* gene expression. The predicted number of fragments per kilobase transcript per million mapped reads was used to compute the *PmLBD* gene’s expression level (FPKM). Log2 (FPKM+1) values were used to create heat maps, which were then evaluated at the row level.

### 4.5. Plant Material, RNA Extraction and qPCR-Based Analysis

RNA was extracted from various tissues of 15-year-old *P. massoniana* and from 2-year-old seedlings under ten treatments in order to study the expression level of the *PmLBD* gene. Seedlings were transplanted into pots with a peat: perlite: vermiculite (peat: perlite: vermiculite, 3:1:1) soil combination at 24 °C and 16 h of light/8 h of dark light. Seedlings at a comparable stage of growth were chosen for subsequent treatments. Mechanical injury, 10 mM H_2_O_2_, 10 mM methyl jasmonate (MeJA) [43], 50 mM ethylene glycol (ETH), 1 mM salicylic acid (SA), 15% PEG6000 [47], 100 mM abscisic acid (ABA) [33], 150 mM NaCl [48], 10 M IAA, and 2 mM GAs were the ten treatments employed in this investigation. Leaf samples were utilized for RNA extraction. Afterwards, seedlings were treated for 0, 3, 6, 12, and 24 h. As controls, 0 h samples were used. Each treatment was carried out three times independently as biological duplicates. Prior to RNA isolation, all obtained samples were promptly flash-frozen in liquid nitrogen and kept at −80 °C. Following the manufacturer’s recommendations, total RNA was extracted using the DP441 RNAPEP Pure Kit (Tiangen Biotechnology, Beijing, China) in conjunction with a gDNA removal step. RNA concentration and purity were determined using a NanoDrop 2000 device (Termo Fisher Scientific, Waltham, MA, USA), and RNA integrity was determined using 1.2% agarose gel electrophoresis. A First Strand cDNA Synthesis Kit (11141, Yeasen Biotech, Shanghai, China) was used to create cDNA (20 µL) from 1000 ng of total RNA as directed by the manufacturer. SYBR Green Real-time PCR Master Mix was used to run quantitative PCR on the StepOne Plus equipment (Foster City Applied Biosystems, Foster City, CA, USA) (QPK-201, Toyobo Bio Technology, Shanghai, China). It was decided to use the internal control gene -Microtubulin (TUA) [49]. Appendix A lists the gene-specific primers used for qPCR. Each PCR mixture (10 µL) contained 0.4 µL of each primer (10 mM), 3.2 µL of ddH2O, 5 µL of SYBR Green real-time PCR master mix, and 1 µL of diluted cDNA (20-fold dilution). Pre-denaturation at 95 °C for two minutes, denaturation at 95 °C for forty cycles of ten seconds, and annealing and extension at 60 °C for thirty seconds were the amplification conditions. Melting curves were analyzed from 60 to 95 °C to confirm primer specificity and the lack of primer dimerization. Each reaction was repeated three times. Negative controls were included on each plate and each sample, and cDNA was replaced with ddH_2_O and total RNA. Relative expression levels were calculated according to the 2^−ΔΔCt^ method [50]. All of the above were analyzed using one-way ANOVA and multiple comparison tests on the data were obtained.

### 4.6. Decoy Vector Construction and Self-Activation Detection

Primers were designed according to the sequences of the four selected *PmLBD* genes as shown in Appendix A. *Escherichia coli* solutes were extracted for PCR amplification and purified for recovery. The pGBKT7 vector was ligated with the recovered PCR products after double digestion with BamHⅠ and EcoRⅠ. The ligated products were transformed into a DH5α *E. coli* receptor state and incubated at 37 °C in an inverted position. Single colonies were picked for PCR, agarose gel electrophoresis was used, and the correctly detected bacterial broth was sent to Biocom for sequencing. The correctly sequenced pGBKT7-PmLBD recombinant plasmid was extracted and transformed into the AH109 yeast receptor state. Single colonies of successfully transformed yeast were picked, cultured in a SD/-Trp liquid medium and incubated overnight at 30 °C. 10 µL of yeast dilution was spotted onto SD/-Trp, SD/-Trp/-His/-Ade/X-α-Gal solid medium and incubated upside down at 30 °C. Yeast growth and growth status were observed and recorded [51].

## 5. Conclusions

The identification and classification of LBD transcription factor families is essential in the functional studies of *LBD* genes in *P**. massoniana*. However, there are few reports in this area. To complement this aspect of research, we identified and classified the *PmLBD* families for the first time. The results showed that 47 LBD predicted proteins were identified from four *P. massoniana* transcriptomes. Subcellular localization was predicted to be at the nucleus, and *PmLBD44* and *PmLBD45* were selected for subcellular localization, which showed localization at the nucleus. The transcriptional self-activation assay showed that *PmLBD44*, *PmLBD45*, *PmLBD46* and *PmLBD47* had no transcriptional self-activation activity. The qPCR experiments were performed on seedlings of *P. massoniana* under different stress, injury and phytohormone treatments at different time periods, and the expression of the genes was found to be increased or down-regulated to some extent under different treatments, indicating that this gene family may play a positive response in the process of plant growth and development, abiotic stress and pest and disease attack. The *PmLBD* genes are candidates for further functional analysis and have great potential in the response to abiotic stresses in *P. massoniana*.

## Figures and Tables

**Figure 1 ijms-23-13215-f001:**
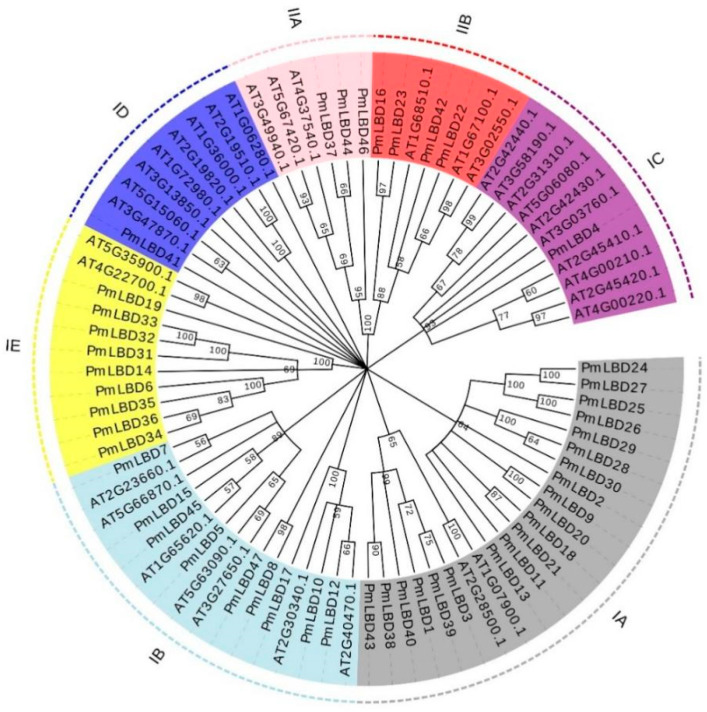
Phylogenetic grouping of members of the *LBD* family of *P. massoniana*. Gene names of different colors correspond to the groups of the same color in the outermost circle, and the numbers on the nodes in the figure indicate branch support values.

**Figure 2 ijms-23-13215-f002:**
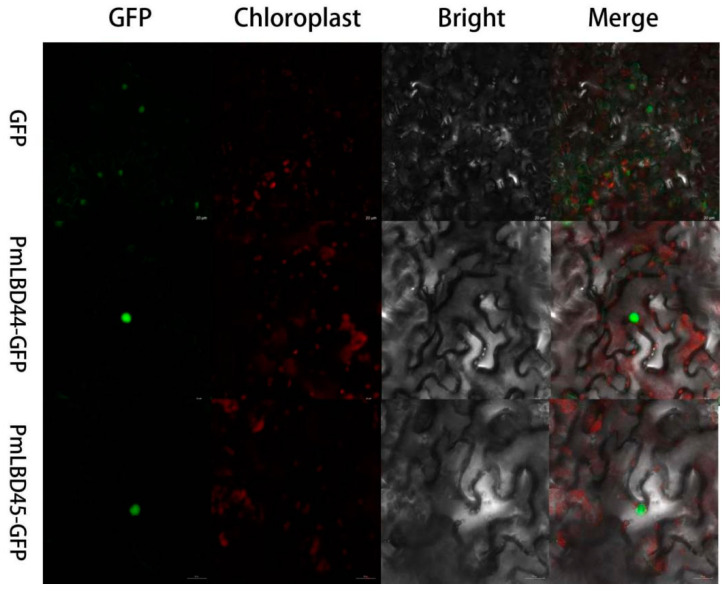
Subcellular localization map of *PmLBD44* and *PmLBD45* in leaves of *Nicotiana benthamiana*. Empty 121-GFP is localized in the whole cell, and 121-PmLBD-GFP is localized in the nucleus. Scale bar = 20 µm.

**Figure 3 ijms-23-13215-f003:**
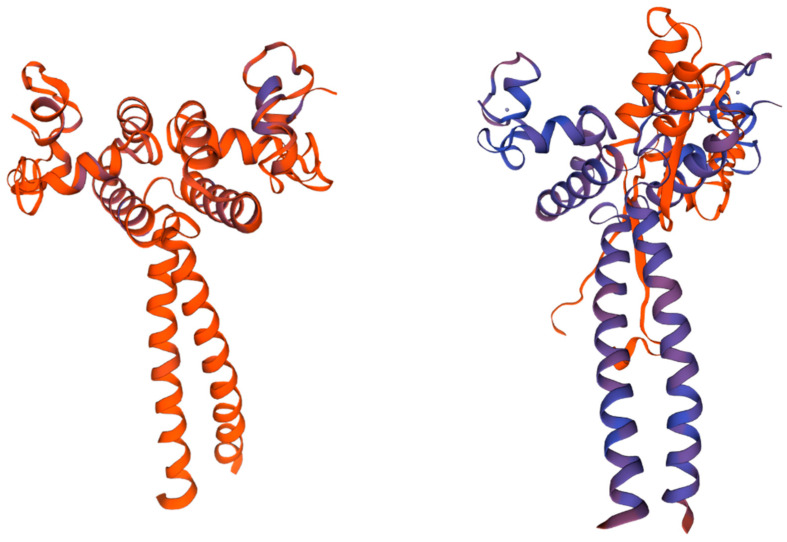
Tertiary structure of horsetail pine LBD protein. Model with multiple α-helices, less irregular curl and β-turn, no β-folding found. (left: *PmLBD44* vs. right: *PmLBD47*).

**Figure 4 ijms-23-13215-f004:**
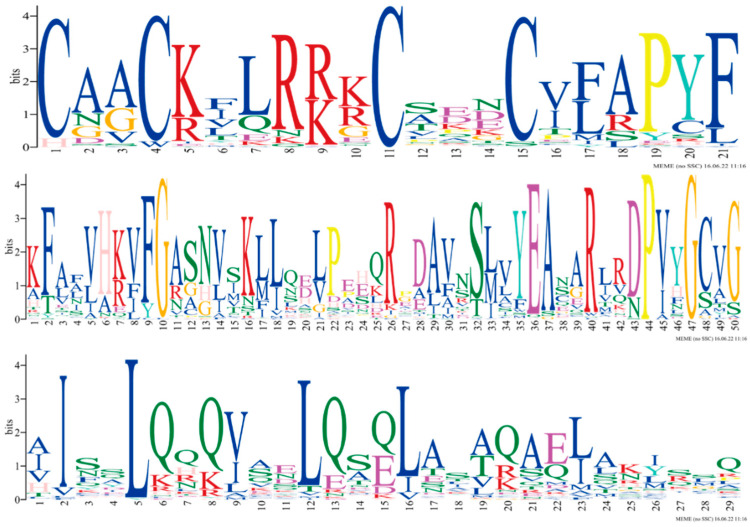
Sequence signature of LOB structural domain extracted from LBD protein sequence information. The motif1 is the zinc finger-like structure CX2CX6CX3C, motif2 is the GAS motif, and motif3 is the leucine-like zipper structure LX6LX3LX6L.

**Figure 5 ijms-23-13215-f005:**
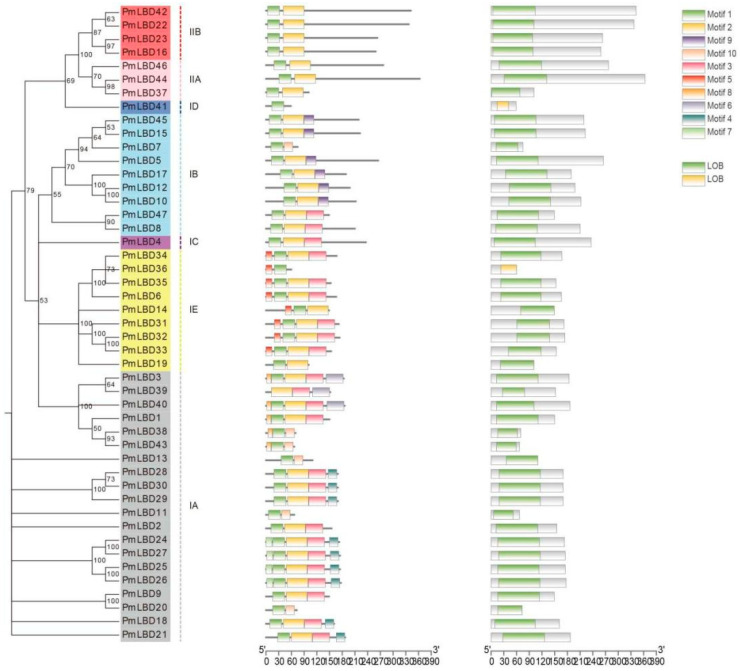
Phylogenetic grouping, motif distribution and structural domain analysis of LBD proteins in *P*. *massoniana*.

**Figure 6 ijms-23-13215-f006:**
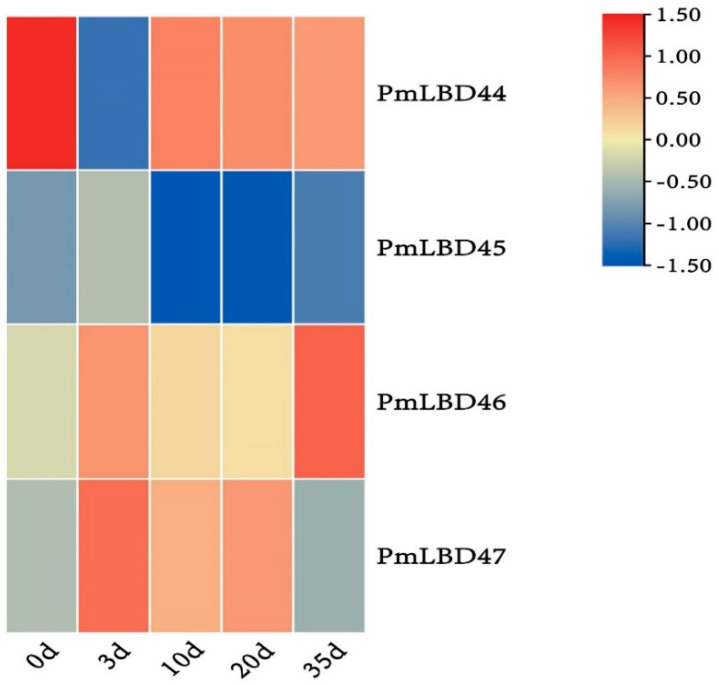
Transcription products of the *LBD* gene at different stages during inoculation with *P. massoniana* nematodes: days 0, 3, 10, 20 and 35. Heat maps were generated using log2 (FPKM+1) values and then line scales were executed. Dark blue indicates low expression levels, light colors indicate moderate expression levels, and red indicates high expression levels.

**Figure 7 ijms-23-13215-f007:**
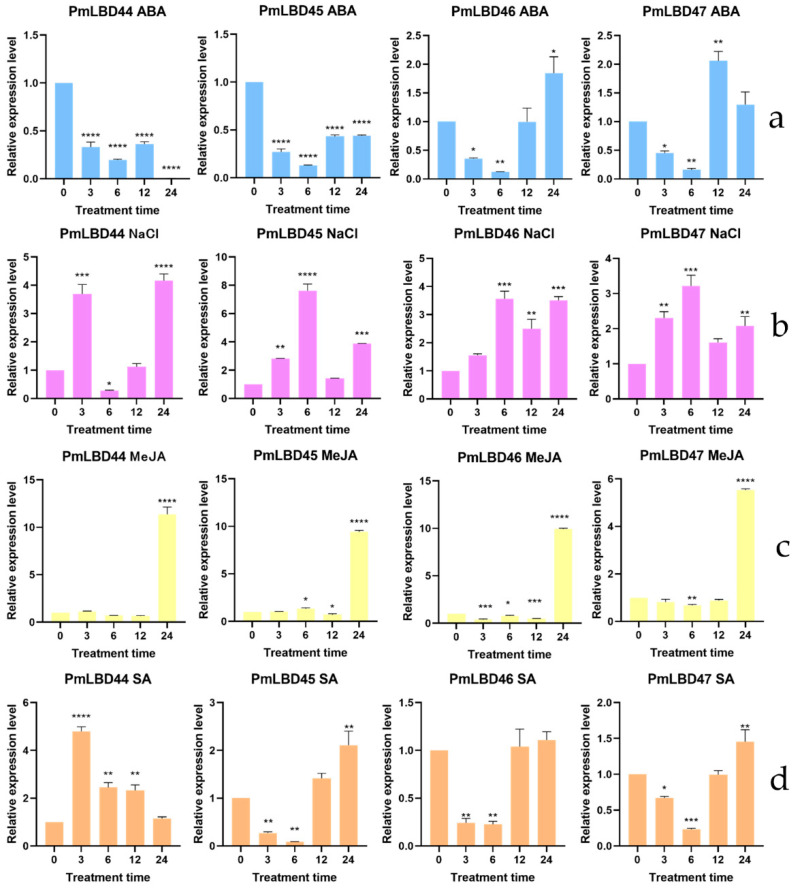
Effect of different treatments associated with pine nematode infection on *LBD* gene expression. (**a**) ABA, (**b**) NaCl, (**c**) MeJA, (**d**) SA, (**e**) IAA, (**f**) PEG, (**g**) Mechanical Injury, (**h**) H_2_O_2_, (**i**) ETH. Asterisks indicate significant differences in transcript abundance in the treated group compared to the control group (0 h) (* *p* < 0.05, ** *p* < 0.01, *** *p* < 0.001, **** *p* < 0.0001).

**Figure 8 ijms-23-13215-f008:**
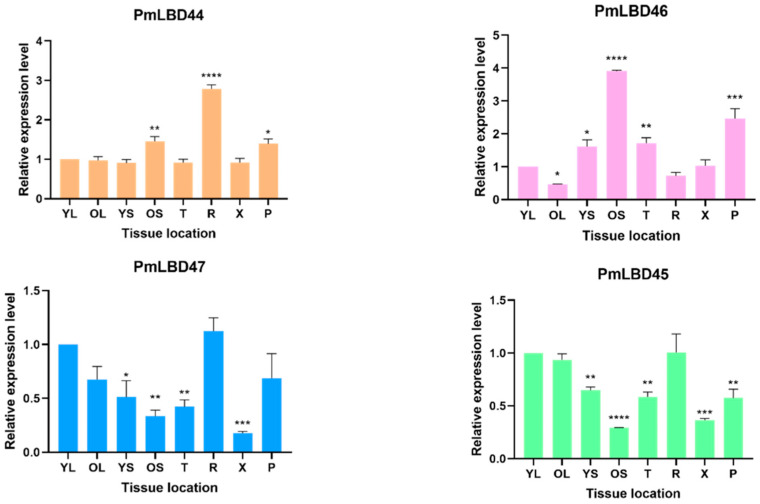
Analysis of *LBD* gene expression under different tissue sites. The asterisks indicate significant differences in transcript abundance in the treatment group compared to the control group (YL) (* *p* < 0.05, ** *p* < 0.01, *** *p* < 0.001, **** *p* < 0.0001). yl: young leaf; ol: old leaf; ys: young stem; os: old stem; t: apical; r: root; x: xylem; p: bast.

**Figure 9 ijms-23-13215-f009:**
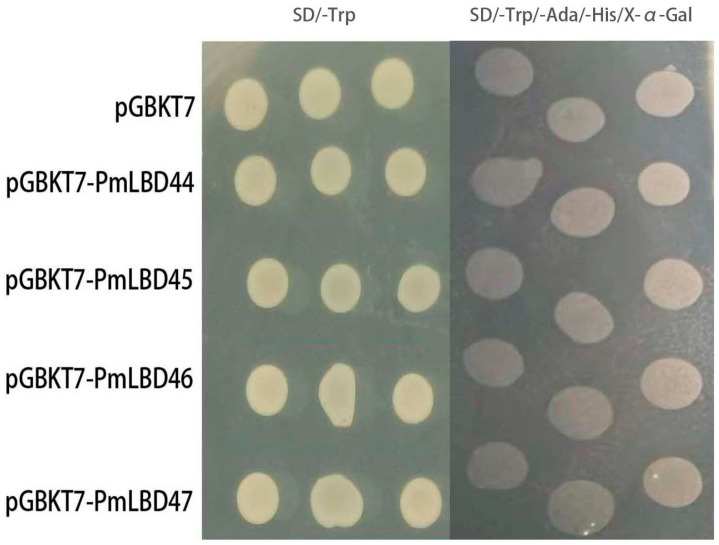
Transcriptional self-activation assay for pGBKT7-PmLBD vector. The negative control pGBKT7 was incubated empty with the decoy vector pGBKT7-PmLBD co-spotted with SD/-Trp and SD/-Trp/-His/-Ada/X-α-gal, and both were found to grow only on SD/-Trp plates.

**Table 1 ijms-23-13215-t001:** Secondary structure analysis of the *PmLBD* gene family.

Name	Alpha-Helix	Beta-Folding	Irregular Curl
*PmLBD1*	60.67	1.33	36
*PmLBD2*	51.95	0.65	38.96
*PmLBD3*	54.1	3.83	34.97
*PmLBD4*	37.45	3.83	52.34
*PmLBD5*	44.32	3.79	45.45
*PmLBD6*	60	3.64	30.91
*PmLBD7*	43.24	2.7	48.65
*PmLBD8*	64.11	0.48	33.49
*PmLBD9*	58.78	0	35.14
*PmLBD10*	51.66	0.47	42.65
*PmLBD11*	27.27	9.09	45.45
*PmLBD12*	52.79	3.05	37.56
*PmLBD13*	42.73	4.55	38.18
*PmLBD14*	48.99	5.37	34.9
*PmLBD15*	51.58	2.71	41.18
*PmLBD16*	22.87	5.43	58.91
*PmLBD17*	42.55	1.06	55.32
*PmLBD18*	58.75	1.88	35.62
*PmLBD19*	40.59	0	55.45
*PmLBD20*	42.47	2.74	49.32
*PmLBD21*	56.45	3.23	33.33
*PmLBD22*	34.52	6.55	43.15
*PmLBD23*	27.86	7.63	49.62
*PmLBD24*	48.26	1.74	44.19
*PmLBD25*	50	0	44.25
*PmLBD26*	52.27	0.57	42.61
*PmLBD27*	50.57	0.57	41.38
*PmLBD28*	52.07	0.59	43.2
*PmLBD29*	57.99	1.18	37.87
*PmLBD30*	52.07	1.18	40.83
*PmLBD31*	60.82	0.58	33.33
*PmLBD32*	60.12	8.09	25.43
*PmLBD33*	60.78	7.84	27.45
*PmLBD34*	66.27	5.42	27.11
*PmLBD35*	61.84	5.92	30.26
*PmLBD36*	51.67	5	36.67
*PmLBD37*	31	13	31
*PmLBD38*	44.93	5.8	39.13
*PmLBD39*	56.95	3.31	37.09
*PmLBD40*	56.76	5.95	26.49
*PmLBD41*	24.14	0	58.62
*PmLBD42*	34.9	6.45	44.87
*PmLBD43*	43.94	4.55	48.48
*PmLBD44*	25.69	8.56	42.82
*PmLBD46*	27.9	5.8	52.9
*PmLBD47*	49.32	0	47.97

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
