# Peer review of "Identification, Classification and Characterization of LBD Transcription Factor Family Genes in *Pinus massoniana"

_ijms, 2022, doi:10.3390/ijms232113215_

Round 1
Reviewer 1 Report
The identification and classification of LBD transcription factor families is essential in the functional studies of LBD genes in P. massoniana.Does not fully address the role of TF in resistance to resistance to the nematode. The article contains incorrect data.
For example, there is a lack of data on the molecular and genetic characteristics of Pinaceae species and limited access to genomic information.
Reviewer 2 Report
The manuscript entitled “Identification, classification and characterization of LBD transcription factor family genes in Pinus massoniana” described the identification of 47 LBD gene family members in the confier, Pinus massoniana. The authors performed gene expression study to understand the roles of LBD genes in various tissues and during stress treatments. Sub-cellular localization of couple of genes were also presented. Although the manuscript is interesting, I have following concerns regarding various experiments carried out in the study.
1. Section 3.2.2, Authors performed localization studies of pmLBD44 and pmLBD45 genes tagged with GFP and concluded that the proteins are localized to nucleus. The experiment was carried out without proper controls. Authors should use nucleus marker tagged with fluorescent tag as control and then provide an overlay image. The data provide is not strong enough to support the author claims of its nuclear localization.
2. Authors should validate the RNA-seq data by q-PCR and then should provide an overlay graph of both the results.
3. Authors should represent Genes names in italics and protein names in non-italics.
4. In Figure 7, graph panel 7 needs translation from Chinese to English.
5. In all the figures authors should define p value for *** and also provide the statistical method used to define significant difference.
Reviewer 3 Report
The paper "Identification, classification and characterization of LBD tran-scription factor family genes in Pinus massoniana" by Zhang et al. is mainly based on a bioinformatic study employing transcriptional datatabases to identify and characterize the LBD protein family in Pinus massoniana. The authors add some experimental approaches to confirm bioinformatic data and obtain some information about regulation of LBD by growth fantors and mainly by biotic or abiotic stresses. This study seems quite complete, although accurate functional conclusions fail. There are several flaws and points that might be reviewed and corrected:
The legend for Fig. 2 does not explain the results presented in the microphotagraphs. Here and in the main text explanations must be completed. The same for Figs. 3, 4 and 9.
In point 3.4 must be indicated the origin of transcriptomic data and a reference should be required.
Point 3.5: Phrasing and punctuation must be reviewed.
Point 3.7: The meaning of "carrier toxicity" and "self activating activity" must be explained. Also must be explained more clearly what is shown in Fig. 9.
Discussion: The following sentence is confusing and speculative (cannot deduced from results presented in this paper): "Changes in gene expression in response to treatment with the growth hormone IAA suggest that reduced growth hormone accumulation may regulate plant growth adaptation by regulating growth hormone receptor ex-pression maintaining a low growth hormone signaling response"
The functional sense of "it was demonstrated that PmLBD44, PmLBD45, PmLBD46 and PmLBD47 did not possess transcriptional self-activating activity" must be explained.
Conclusions:
The sentence "Based on the qPCR data information, we found that the expression levels of PmLBD genes under different treatments indicated that they were essential for the growth and development of P. massoniana..." is not supported by the results.
Round 2
Reviewer 2 Report
Author’s addressed all my concerns and the article can be accepted